# Spoilage Monitoring and Early Warning for Apples in Storage Using Gas Sensors and Chemometrics

**DOI:** 10.3390/foods12152968

**Published:** 2023-08-06

**Authors:** Limei Yin, Heera Jayan, Jianrong Cai, Hesham R. El-Seedi, Zhiming Guo, Xiaobo Zou

**Affiliations:** 1Key Laboratory of Modern Agricultural Equipment and Technology, Ministry of Education, School of Agricultural Engineering, Jiangsu University, Zhenjiang 212013, China; yinlm6@163.com; 2School of Food and Biological Engineering, Jiangsu University, Zhenjiang 212013, China; heerajayan93@outlook.com (H.J.); jrcai@ujs.edu.cn (J.C.); zou_xiaobo@ujs.edu.cn (X.Z.); 3Pharmacognosy Group, Department of Pharmaceutical Biosciences, Biology Medical Center, Uppsala University, P.O. Box 591, SE-751 24 Uppsala, Sweden; hesham.el-seedi@fkog.uu.se; 4International Joint Research Laboratory of Intelligent Agriculture and Agri-Products Processing, Jiangsu University, Zhenjiang 212013, China

**Keywords:** gas sensor, spoilage monitoring, early warning, logistics control, simulated annealing, apple

## Abstract

In the process of storage and cold chain logistics, apples are prone to physical bumps or microbial infection, which easily leads to spoilage in the micro-environment, resulting in widespread infection and serious post-harvest economic losses. Thus, development of methods for monitoring apple spoilage and providing early warning of spoilage has become the focus for post-harvest loss reduction. Thus, in this study, a spoilage monitoring and early warning system was developed by measuring volatile component production during apple spoilage combined with chemometric analysis. An apple spoilage monitoring prototype was designed to include a gas monitoring array capable of measuring volatile organic compounds, such as CO_2_, O_2_ and C_2_H_4_, integrated with the temperature and humidity sensor. The sensor information from a simulated apple warehouse was obtained by the prototype, and a multi-factor fusion early warning model of apple spoilage was established based on various modeling methods. Simulated annealing–partial least squares (SA-PLS) was the optimal model with the correlation coefficient of prediction set (R_p_) and root mean square error of prediction (RMSEP) of 0.936 and 0.828, respectively. The real-time evaluation of the spoilage was successfully obtained by loading an optimal monitoring and warning model into the microcontroller. An apple remote monitoring and early warning platform was built to visualize the apple warehouse’s sensors data and spoilage level. The results demonstrated that the prototype based on characteristic gas sensor array could effectively monitor and warn apple spoilage.

## 1. Introduction

Apple has the characteristics of high nutritional value and storage resistance, but due to its high sugar and moisture, coupled with complex external environmental factors, it is susceptible to fungal spoilage [1,2]. Apple spoilage is the result of changes in tissue composition under the action of physical, chemical, microbial and other environmental factors. Physical aspects include mechanical damage, frostbite, etc.; chemical aspects include water imbalance and quality deterioration caused by environmental changes; microbiological aspects include fruit rot, penicillium and rot heart disease, etc. Among them, the spoilage caused by fungi is the most serious. With the increase in storage time, spoilage fungi on the surface of apples enter the fruit through stomata or calyx and other parts to generate mycelium, which eventually leads to the spoilage of apples. In addition, spoilage fungi may also produce mycotoxins that pose the risk of disease, which can seriously endanger the health of consumers [3]. By sampling and isolating the epiphytic microbiota of fresh apples, it was found that the proportion of fungi (79.0%) was much higher than that of bacteria (13.8%) [4], and the dominant spoilage fungi included *Aspergillus niger* (*A. niger*), *Penicillium expansum*, (*P. expansum*), *Penicillium chrysogenum* (*Penicillium chrysogenum*, *P. chrysogenum*) and *Alternaria alternata* (*A. alternata*), etc. [5,6,7,8]. Therefore, monitoring and providing early warning of apple spoilage have important practical significance, and can ensure the food safety of consumers and provide technical support for the healthy development of the apple industry.

At present, the traditional research on the detection of fruit and vegetable spoilage mainly includes polymerase chain reaction (PCR) [9] and gas chromatography-mass spectrometry (GC-MS) [10]. Although these detection methods are accurate, the operation is complicated and requires professional operators, which cannot meet the needs of rapid real-time detection of apple spoilage [11]. Therefore, it is of practical significance to explore a fast and effective apple spoilage monitoring technique.

Electronic nose technology is a powerful tool that simulates the olfactory system of animals [12]. The electronic nose converts chemical signals into electrical signals through the gas sensor array and combines chemometrics to process the data matrix to realize the qualitative and quantitative analysis of the detected samples. It integrates sensors, computers, mathematics and other disciplines, and has been widely used in food fields, such as medicine and the environment [13]. The aroma of fruit is an important indicator for evaluating fruit quality, and is mainly composed of various volatile components. The type and concentration of volatile components can be affected by a variety of factors, such as actual type, maturity and storage time. Electronic noses are widely used in the field of fruit and vegetable testing. By acquiring the gas data of the testing samples, non-destructive rapid testing of the quality of fruits and vegetables can be achieved.

With the development of sensor technology and chemometrics, the accuracy of gas detection has gradually improved, and electronic nose technology has been gradually applied in the field of fruit and vegetable spoilage detection [14]. When the internal quality of fruits and vegetables changes or spoilage occurs, the volatile gases of fruits and vegetables change accordingly. By analyzing the type and content of volatile gases in fruits and vegetables, the degree of spoilage of fruits and vegetables and the detection of spoilage microorganisms can be performed.

Nouri et al. [15] took pomegranate as the research object and used the electronic nose system to identify pomegranates infected with Alternaria, with an accuracy rate of 100%, and reviewed the application of gas sensors in the detection of pomegranates infected with *Alternaria*. Liu et al. [16] proposed a non-destructive testing method based on hyperspectral imaging and electronic nose, which could rapidly detect the microbial content and variety of attributes during the rotting process of strawberries. This research used PCA to extract feature information and established a quantitative prediction model for strawberry microbial content and quality traits, indicating that the changes in the appearance and internal components of fungal-infected strawberries during storage were highly correlated with microbial content. Previous studies have shown that the combination of hyperspectral imaging and electronic nose can help improve the evaluation of strawberry quality and safety.

The spoilage of apples is caused by physiological disorders or aging of apple tissues, and infection by decay-causing microorganisms. The respiration of apples and the catabolism of microorganisms will change the composition and proportion of gases in the storage micro-environment. By clarifying the evolution law of gas composition in an apple spoilage environment, real-time monitoring of indicative gas in the storage environment is an effective way to provide early warning of apple spoilage. Electronic noses are currently widely used in the field of gas monitoring; however, the information collected by traditional desktop electronic noses cannot be linked to the spoilage status of apples, and most of the research objects are single apple samples, and thus the data cannot be applied to a monitoring and early warning model for the whole environment of fruit warehouse. Therefore, this study designed and developed a prototype of apple spoilage storage monitoring and attempted to analyze the change law of gas in the process of batch apple spoilage and the spoilage influence mechanism of multi-factor coupling. An apple spoilage early warning model with various influencing factors was established, and at the same time, a remote monitoring and early warning platform was built to realize remote monitoring of spoilage early warning information.

## 2. Materials and Methods

### 2.1. Design of Monitoring Prototype

The monitoring prototype was designed to detect volatile profiles, temperature and humidity during apple storage. The prototype consisted of hardware, software and mechanical components. The hardware system measured the gas components and included a microcontroller, gas sensor array and display screen. The software system was developed to control the microcontroller and to interface with the computer. The mechanical components involved a gas delivery system that transported the gas components from the storage environment to the sensors array. The design characteristics of the monitoring prototype are further explained in the following sections.

#### 2.1.1. Selection and Optimization of Sensors

The complexity of gas composition in the apple storage environment was fully considered in the selection of sensors. The current storage method in the warehouse mainly adopts a combination of controlled atmosphere and refrigeration, and the storage gas concentration changes in real time. The high sensitivity of the gas sensor helps in the early detection of apple spoilage. Meanwhile, the field layout prototype needs to be small and high-precision with low power consumption and high accuracy to meet the needs of spoilage monitoring. After extensive research and experiments, the main gases in the apple quality and spoilage process were determined, and C_2_H_4_, CO_2_, volatile organic compounds (VOC) and O_2_ were optimized to be the characteristic gases of the apple warehouse [17,18,19,20]. An infrared gas sensor was selected for the CO_2_ sensor, and the remaining sensors were electrochemical gas sensors. This ensured that the prototype had low power consumption and high precision, which is convenient for long-term monitoring in warehouses [21]. Table 1 shows the detection range, resolution, sampling accuracy and repeatability parameters of each sensor.

#### 2.1.2. Air Chamber and Air Path Design

The design requirements of the prototype need to ensure the portability of the equipment. To improve the speed and efficiency of the gas contact sensor surface, the design of the gas chamber is considered, with factors such as the size, structure, and material of the gas chamber [22]. By designing the shell with air holes, the gas sensor array was wrapped, and each gas sensor scatter arranged at the bottom. In order to ensure the circulation of the gas flow path, the external gas was sucked into the prototype by the fan and evenly passed through the surface of the sensor array to better obtain the gas information of the storage micro-environment.

#### 2.1.3. Hardware System Integration

The hardware system included a microcontroller, gas sensor array, TF card module, micro vacuum pump, power supply and display screen [23]. Figure 1 shows the schematic representation of various components in the prototype. The microcontroller was mainly used to control the collection of gas sensor data and control other hardware devices, such as fans. The gas sensor array was used to obtain the data of the gas in the apple warehouse. The TF card module was used to store the gas data. The display screen revealed the real-time sensing data of each gas sensor.

The miniature air pump drove the flow of gas in the air chamber, and the one-way valve controlled the closure of the air path. The power module supplied power to the prototype, and the voltage was stabilized to 5V through the voltage regulator circuit to supply power to each sensor.

#### 2.1.4. Software Structure Design

The software was developed under the Windows 10 operating system. The Windows operating system has been affirmed and welcomed by consumers and developers and has now been released to Windows 11. The Windows system provides many development interfaces and standards, and the maintenance difficulty is lower than other systems. The Windows 10 Professional operating system used in this study was based on the NT core, with good hardware support and higher development efficiency.

The microcontroller program was written in Keil uVision5 IDE using C language. Keil provides many library functions and development and debugging tools through the integrated environment, which is convenient for developers to call, and is currently the most popular microcontroller development tool.

Qt is a cross-platform C++ graphical user interface application development framework that enables rapid development of GUI programs and non-GUI programs. Through the visual graphical interface editor, the user can quickly and easily drag and drop controls, including buttons, radio boxes, check boxes, group boxes, tree views, table views and texts. Qt has the advantages of being cross-platform, object-oriented, easy to use and fast to run, and it is easy to transplant and can be quickly converted according to the operating system. It is widely used in the development of embedded products and device interfaces.

The dedicated prototype mainly included sensor signal acquisition, data display and data storage functions. To visualize the monitoring process, the prototype developed a special human–computer interaction interface, and the display of each sensor’s data was mainly realized by the serial port screen. In order to visually display the data of each sensor, a dedicated display interface was designed.

#### 2.1.5. Prototype System Integration

According to the software and hardware design scheme of the above-mentioned special-purpose prototype, the hardware and software systems were integrated, and the prototype assembly was finally completed. After debugging and optimization of the prototype, the repeatability and stability of the prototype were verified by acquiring apple sample information from the warehouse, and the batch test was carried out after reaching the expectation.

### 2.2. Apple Sample Preparation

#### 2.2.1. Activated Culture and Inoculation of Spoilage Fungi

*Aspergillus niger* (CICC2089), the dominant spoilage fungi of apple, was purchased from China Industrial Microbial Species Preservation and Administration Center (CICC). Activation and culture procedures were performed in strict accordance with CICC instructions and guidelines.

The bacterial cells were recovered from lyophilization prior to inoculation. The top of the lyophilized tube with *Aspergillus niger* was placed on the alcohol lamp and heated evenly for 30 s. Then, 2–3 drops of sterile water were dropped onto the heated part. The tube wall was broken due to uneven heat, and the tear was knocked out with sterilized tweezers. The lyophilized powder was placed into a 1.5 mL centrifuge tube using an inoculum ring, and 200 μL of sterile water was added to dissolve it. The lyophilized powder solution was evenly coated on potato dextrose agar medium plates and placed in a constant temperature and humidity incubator at 28 °C. After seven days of culture, the spores of the third generation of fungi were scraped with a one-time inoculation ring and placed in sterile water, which was configured into fungal suspension. The fungal suspension was counted through a blood count plate and diluted with sterile water to a concentration of 10^6^ cfu/mL.

Before inoculation, the apple skin was washed with distilled water, then wiped with 75% alcohol, and finally placed on a sterile workbench under ultraviolet light for half an hour. Apple samples were punctured with sterile syringe needles (diameter 3 mm, depth 5 mm) along the apple equator, with 3 puncture holes, each 120° apart. Then, 5 μL of fungal suspension was injected into each of the three holes and incubated in a constant temperature and humidity incubator (25 °C, 60% humidity) [5].

#### 2.2.2. Micro-Environment Information Sensing

To simulate the conditions of apple storage in warehouses, nine simulated warehouses were set up in the laboratory [24]. Each simulated warehouse contained 30 fresh apple samples, and the gas sensing data and temperature and humidity data of the apple samples were collected for two days by the acquisition terminal prototype. Then, 10 apple samples were selected from each simulated warehouse to be inoculated with *Aspergillus niger*, and the inoculated apple samples were put back into the simulated warehouse. Data acquisitions were performed every 24 h for a total of 6 days. The data format detection system was stored in a two-dimensional table format. The collection time of a single sensor was 500 s, and the collection frequency was 1 s. The data of each simulated warehouse sample were collected as a 500 × 6 two-dimensional array based on 6 sensors. Then, the data were transformed from a 500 × 6 two-dimensional matrix into a 3000 × 1 one-dimensional array through flattening processing for subsequent model establishment.

### 2.3. Variable Selection Method

#### 2.3.1. Genetic Algorithm

Genetic algorithm (GA) is an algorithm based on biological evolution rules, which automatically obtains and guides the optimized search space by simulating random search and optimization solving methods [25], which can quickly screen characteristic variables and eliminate the interference of irrelevant information [26,27], has the characteristics of simple operation and strong versatility, achieves the global optimum in a short time and can reduce the risk of falling into the local optimum search.

#### 2.3.2. Simulated Annealing Algorithm

Simulated annealing (SA) is a probabilistic optimization algorithm for simulating the solid annealing process in metalwork [28]. SA has strict convergence characteristics following a Metropolis criterion, which effectively reduces the probability of falling into a local minimum. SA can quickly find the global optimal solution, and the final optimization result has nothing to do with the initial value [29]. It is a powerful tool for solving optimization and combination problems. SA has the characteristics of simplicity, flexibility and efficiency, which can effectively improve the generalization ability of the model.

#### 2.3.3. Ant Colony Optimization Algorithm

Ant colony optimization (ACO) algorithm was inspired by the bionic intelligence of ant colony foraging behavior [28,30]. Ants use shared pheromones to quickly spread information in ant colonies, which helps to strengthen cooperation between ant colonies, improve global exploration capabilities and obtain better solution results [31]. The essence of ACO is based on its ability to optimize the creation of paths, and it has strong generality and robustness and is widely used in data optimization and fuzzy modeling.

#### 2.3.4. Competitive Adaptive Reweighed Sampling

Competitive adaptive reweighed sampling (CARS) is a variable selection method suitable for high-dimensional data extraction [32]. In the sampling stage, CARS regards each variable as an independent individual, retains variables with larger weights, removes variables with smaller weights, and treats variables with significant weights as a new subset, which can effectively remove irrelevant variables and reduce collinear variables [33,34]. By selecting the optimized subset of variables, the algorithm can overcome the combinatorial explosion problem in variable selection to a certain extent, improve the prediction ability of the model, and reduce the prediction variance. CARS introduces an exponential decay function [35], which controls the retention rate of variables and improves the computational efficiency.

### 2.4. Apple Remote Monitoring and Early Warning Platform

The apple remote monitoring and early warning platform mainly included three parts: data upload module, remote monitoring module and spoilage early warning module, as shown in Figure 2. The platform development language was JAVA, which was developed through the SSM frameworks, including the SpringBoot, SpringMVC and MyBatis 3 frameworks [36,37]. The visualization of individual sensor data and spoilage levels was implemented by the Echarts visualization library [38].

The data upload module was mainly responsible for the upload of sensor and model data, which made it convenient for the subsequent monitoring module and spoilage early warning module to call data. The remote-monitoring module mainly displayed the trend and change of sensor data over time through a line graph and realized the visualization of each sensor’s data. The schematic diagram of apple spoilage monitoring and early warning process is shown in Figure 3. The spoilage early warning module was mainly responsible for calling the data of sensors and spoilage models and realizing the visual display of spoilage levels through the dashboard.

## 3. Results

### 3.1. Analysis of Apple Volatile Gas

The volatile gas production during fungal spoilage of apples was recorded using the sensor and represented as sensor response value. The sensor response value indicated the response value of each sensor to the presence of certain gaseous chemicals over time.

The average response data of the VOC, CO_2_, O_2_ and C_2_H_4_ gas sensors of simulated warehouse apple samples are shown in Figure 4. The response value of the VOC sensor was 7 > 8 > 4 > 6 > 5 > 1 > 3 > 2, and the VOC content on the seventh and eighth days was much higher than that on the first and second days (Figure 4a). The results showed that the content of VOC gradually increased during the degradation of apples from fresh to severe spoilage. The microbial spoilage of apples led to changes in VOC emissions, which are classified as alcohols, terpenes, ketones, alkenes, benzenoids and sulfides [39].

The responses of the CO_2_ sensor indicated that the content on the eighth day was much higher than that on the first day: 1 < 4 < 3 < 2 < 7 < 5 < 6 < 8 in sequence (Figure 4b). The results showed that during the spoilage of apples, the CO_2_ release decreased gradually. The responses of the O_2_ sensor were 8 > 5 > 6 > 7 > 3 > 4 > 2 > 1 in sequence, and the content on the eighth day was much higher than that on the first day (Figure 4c). The results showed that the consumption of O_2_ gradually decreased during apple spoilage process due to the natural senescence process that causes cell and tissue to breakdown.

The responses of the C_2_H_4_ sensor were 7 > 6 > 8 > 5 > 4 > 3 > 2 > 1 in sequence, with the highest content on the seventh day and the lowest content on the first day (Figure 4d). During the decaying process of apples from fresh to severe spoilage, the production of C_2_H_4_ increased gradually, except that the content decreased slightly on the eighth day of treatment [24].

Based on the results obtained [24,40], it was found that with the intensification of the degree of spoilage, the release of VOC and C_2_H_4_ gradually increased, while the release of CO_2_ and the consumption of O_2_ were generally reduced. This was because the metabolic capacity of apples decreases as they spoil, leading to a lower consumption of O_2_ and a lower release of CO_2_. With the increase in spoilage time, the apple samples consume O_2_ and release CO_2_, VOC and C_2_H_4_, but as the degree of spoilage progresses, the metabolic capacity of apples is reduced, leading to the low consumption of O_2_ and the release of CO_2_.

### 3.2. Apple Spoilage Early Warning Model

#### 3.2.1. ACO-PLS Prediction Model of Apple Spoilage

The optimized parameters of the algorithm were set as follows: the initial population size was 50, the maximum number of cycles was 10, the maximum number of iterations was 50, the variable selection probability threshold P was 0.3 and the significance factor Q was 0.01. The characteristic variables of the sensor data were screened by the ACO method, and the screening results are shown in Figure 5a. Forty-eight characteristic variables were screened, and the ACO-PLS model results are shown in Figure 6a.

#### 3.2.2. CARS-PLS Prediction Model of Apple Spoilage

The main parameters of CARS in this study were set as follows: the maximum number of principal components was 15, the number of interactive validation groups was 5 and the number of Monte Carlo sampling runs was 2000. Figure 5b shows the best calculation result of the CARS model of apple spoilage time. It can be seen from the figure that the RMSECV value was large at the beginning of the screening, the regression coefficient of each variable was small, and the number of variables was larger. With the increase in sampling times, the number of variables gradually decreased and the gap between the regression coefficients of each variable widened; 22 variables were screened out. The results of CARS-PLS are shown in Figure 6b.

#### 3.2.3. GA-PLS Prediction Model of Apple Spoilage

The main parameters of GA were set as follows: the number of initial chromosomes was 30, the deletion group was 5, the mutation rate was 0.01, the crossover probability was 0.5 and the maximum number of iterations was set to 100. The cumulative frequency of the variables screened by the GA algorithm is shown in Figure 5c, and 32 variables were screened out. The PLS quantitative prediction model corresponding to the apple spoilage area was established through the screened characteristic variables. The specific scatter diagram of the model is shown in Figure 6c.

#### 3.2.4. SA-PLS Prediction Model of Apple Spoilage

The main parameters of SA in this study were set as follows: the initial temperature was 10, the end temperature was 1, the Markov chain length was 10, the temperature cooling coefficient was 0.95, the window starting width was 10 and the window ending width was 20 and increasing. The step size was 1, the number of wavenumber points exchanged each time the Markov chain was 2 and the maximum number of principal components for modeling was 12. Figure 5d shows the results of variable screening of sensor data through SA, and 20 characteristic variables were screened out. The results of the established SA-PLS model are shown in Figure 6d.

### 3.3. Comparison and Analysis of Various Models

The variable selection method was selected to filter the characteristic variables of the sensor data, and the PLS prediction model of the number of days of apple spoilage was established. The specific results of the apple spoilage time prediction model established by the variable selection method are shown in Table 2. The scatter plot of the apple spoilage time prediction model is shown in Figure 6, in which the R_c_ and R_p_ of ACO-PLS are 0.971 and 0.926, respectively, the R_c_ and R_p_ of SA-PLS are 0.942 and 0.936, respectively, and ACO-PLS has the highest R_c_, but R_p_ is low. In order to ensure the prediction accuracy of the apple spoilage model, the variables were numbered from 1 to 3000, and SA-PLS was used to establish an early warning model of apple spoilage. The characteristic variables were identified to be 1889, 1894, 1974, 2001, 2159, 2163, 2274, 2561, 2758 and 2965. A similar observation was made by Guo et al. [41], where they observed that the competitive adaptive reweighted sampling (CARS) algorithm combined with PLS effectively filtered irrelevant information and improved the accuracy of the model in predicting apple spoilage area from the electronic nose data. Table 3 showed the characteristic variables and original variable ranges screened by SA.

According to the results of the SA-PLS spoilage early warning model, the dependent variables, independent variables, and coefficients of the model were derived. The model results are shown in Table 4. The apple spoilage early warning model is as follows: Y = 0.3264 X1 + 0.3708 X2 + 0.0248 X3 + 0.0363 X4 − 0.0008 X5 − 0.0005 X6 − 0.0014 X7 + 0.4734 X8 + 0.3338 X9 +0.0248 X10 − 0.0136 X11 − 0.0118 X12 − 0.0132 X13 + 0.3407 X14 − 1.9581 X15 + 0.3719 X16 + 0.5173 X17 − 1.9010 X18 + 0.0013 X19 − 0.0009 X20 + 38.9899. Among these values, X1–X20 are the dependent variables, that is, the value of the sensor corresponding to the screening feature variable. When the value of Y is in the range of 1–8, the Y value from 1–2 indicates the freshness of the product, whereas a value between 3 and 4 indicates spoilage. Similarly, a value of 5–6 indicates that the spoilage grade is medium spoilage, and a value of 7–8 indicates that the spoilage grade is severe spoilage.

## 4. Discussion

Apples are prone to spoilage due to physical damage and microbial infections, which can lead to widespread infection and post-harvest losses. The current study has developed a sensor prototype for apple spoilage monitoring and an early warning system for apples. The gas composition information collected by the sensor prototype from the simulated warehouse was used to develop a multi-factor fusion early warning model to predict apple spoilage during storage. Among the different models employed for the purpose, the ACO-PLS model showed the highest correlation coefficient in calibration as well as prediction set. However, it is worth mentioning that the SA-PLS model was suitable for identifying the characteristic variables. A similar outcome was obtained by Ren et al. [38] when a multilayer perceptron neural network (MLPN) model was employed on volatile components released by damaged apples upon mechanical injury. The model was able to classify the degree of damage with 100% accuracy based on the volatile components released from apples. However, the selection of a model for the prediction of the quality of fruits greatly varies depending on the specific situation and intention. Similarly, monitoring gas emitted by fruits and vegetables in combination with multivariate chemometric analysis has been proven to be effective in detecting spoilage inside refrigerators [42]. The use of artificial neural networks as a machine learning model is prevalent in developing a predictive model from electronic nose data [43]. However, compared to neural network models, PLS models are more interpretable, robust to noise and faster to train, making them suitable for real-world applications. With the involvement of too many predictor variables, the PLS model becomes unstable, and the prediction can be inaccurate. Thus, the selection of optimum parameters for modeling is an important task in the process of model building. In this study among the different variable selection methods (ACO, CARS, GA and SA) employed for the reduction of computational complexity, the SA method showed better results. Similarly, Zhao et al. [44] employed SSA to select optimal parameters for the BPNN network model developed for the prediction of fungal infection in apples using electronic nose data.

The developed prototype has a gas monitoring array to collect data on VOC, CO_2_, O_2_, C_2_H_4_, temperature and humidity. These data are then employed to build a multifactor fusion early warning model. Other works have reported that a mathematical model for shelf-life analysis could not be established based only on one specific parameter. The development of the model has shown very high efficiency in establishing shelf-life predictions in several industries, including dairy [45].

The developed model based on the gas sensor prototype has the potential to be employed in the early detection of fruit spoilage, which prevents postharvest losses and reduces economic loss significantly. There were similar attempts to predict the spoilage of bananas based on the 3D fluorescence data of the storage room gas. The spoilage benchmark for the stored banana was estimated as early as the 4th day, which demonstrates the reliability of the early warning systems [46]. However, it is worth mentioning that the early detection system was established based on the fusion of information from no less than five representative physical and chemical indicators from bananas. Moreover, Putnik et al. [47] have developed a mathematical model to compare the influence of modified atmosphere packaging in the prevention of apple browning, and the model was published online as a computer simulation that predicts shelf life given the basic quality parameters of apples, thus proving the practical applicability of model development in predicting apple spoilage. The models can be employed to answer questions about the economic benefits of using different treatments in industrial apple production and storage. This is indirectly helpful in extending the shelf-life of apples and providing economic benefits to producers, while also ensuring that consumers receive high-quality food.

Overall, this work presents a novel approach for monitoring and providing early warning of spoilage of apples in storage based on a gas sensor prototype. The developed system can help reduce post-harvest losses by preventing the spread of infection in the storage environment. Moreover, the developed method can assist in streamlining the process of monitoring and inspecting apples for fungal spoilage.

## 5. Conclusions

In view of the current problems and difficulties of spoilage monitoring in apple warehouses, we analyzed the overall framework of a sensor prototype for the detection of various gases produced during apple spoilage. The prototype was developed by integrating software and hardware development of acquisition terminals to continuously obtain sensor data from simulated warehouses and analyze the process of spoilage in batches of apples. The change in various gaseous composition and their influence on spoilage was identified to be a result of multi-factor coupling. The early warning model for apple spoilage was developed by considering multiple factors, including temperature, humidity and gas composition, and a variable selection method was employed to optimize the characteristic variable, which predicted the degree of spoilage. The developed remote monitoring platform had three major modules, including data upload module, remote monitoring module and spoilage early warning module. These modules enabled the upload of sensor data, visualization of the trends in the data over time and visual display of spoilage levels. Thus, the study proved that analyzing the change in gas composition in an apple spoilage micro-environment and real-time monitoring and warning of indicator gas is an effective way to reduce postharvest losses of fruits.

## Figures and Tables

**Figure 1 foods-12-02968-f001:**
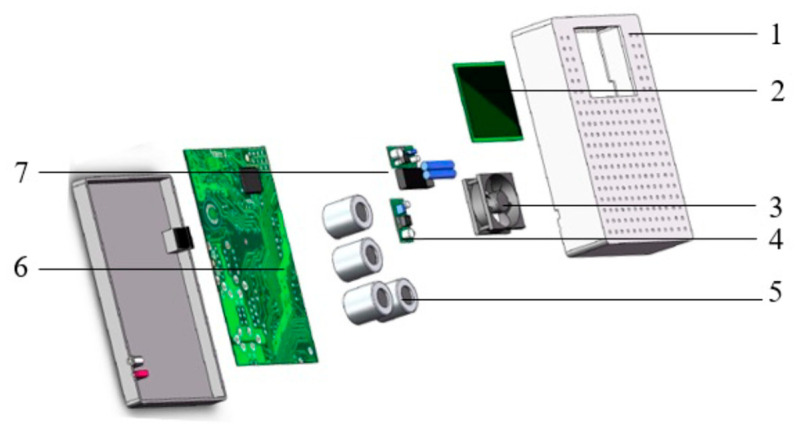
Schematic diagram of the structure of the apple spoilage monitoring prototype.1. Shell. 2. Display. 3. Fan. 4. Temperature/humidity sensor. 5. Gas sensor array. 6. Motherboard. 7. Battery.

**Figure 2 foods-12-02968-f002:**
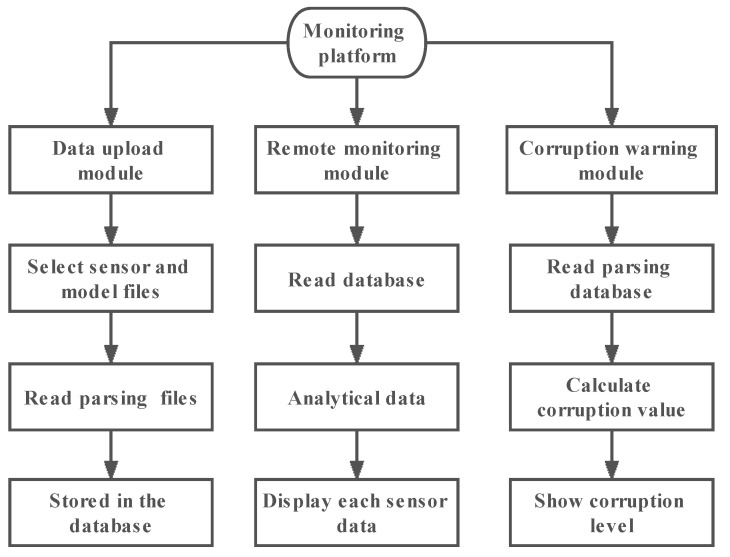
Flow chart of each module of apple remote monitoring and early warning platform.

**Figure 3 foods-12-02968-f003:**
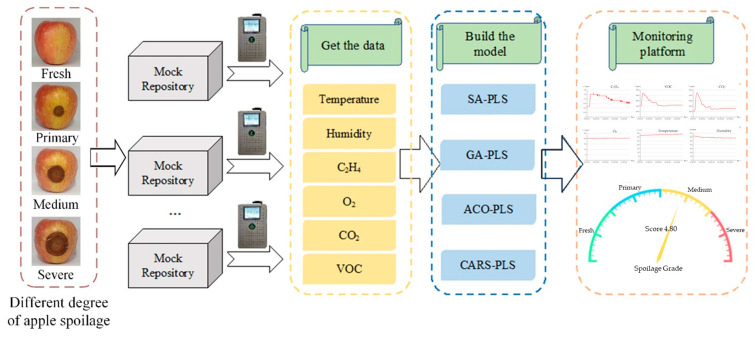
Schematic illustration of apple spoilage early warning model and remote monitoring and early warning.

**Figure 4 foods-12-02968-f004:**
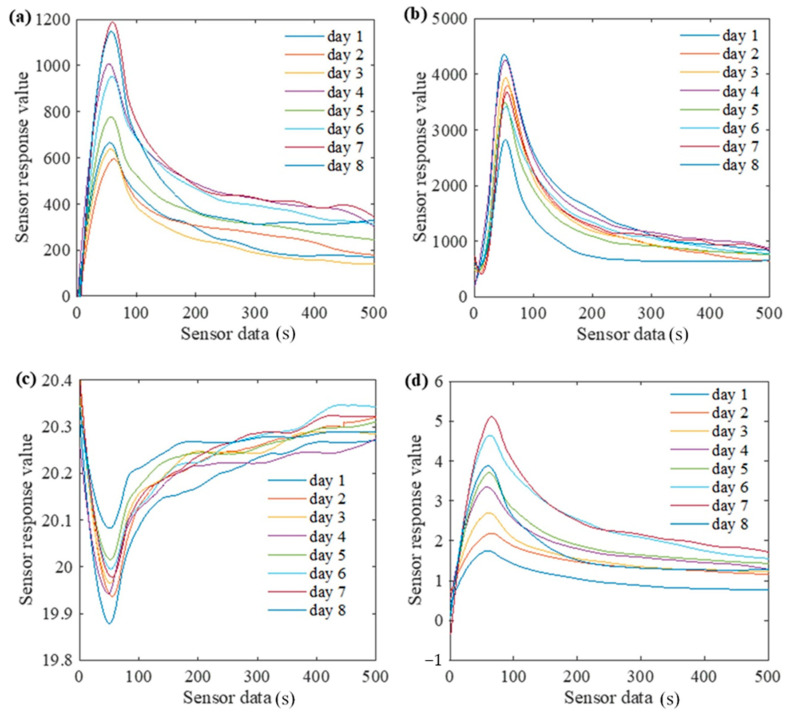
Response of each sensor for different days of spoilage. (**a**) VOC, (**b**) CO_2_, (**c**) O_2_ and (**d**) C_2_H_4_.

**Figure 5 foods-12-02968-f005:**
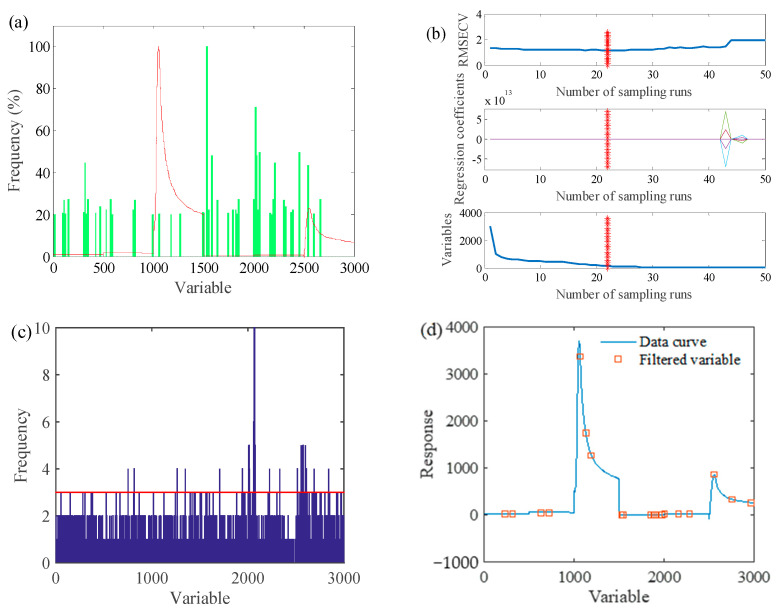
Variable selection results of apple spoilage days. (**a**) ACO-PLS, (**b**) CARS-PLS, (**c**) GA-PLS and (**d**) SA-PLS.

**Figure 6 foods-12-02968-f006:**
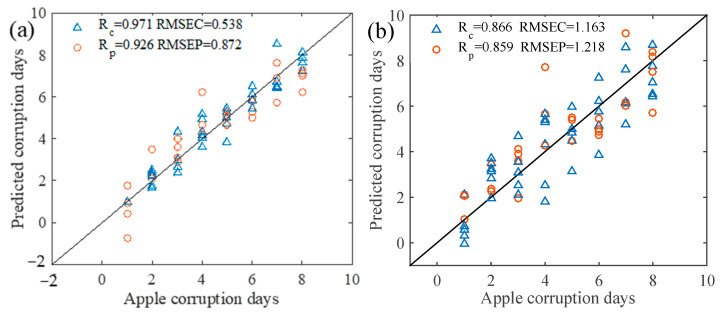
Prediction model of apple decay days. (**a**) ACO-PLS, (**b**) CARS-PLS, (**c**) GA-PLS and (**d**) SA-PLS. Blue triangle represents calibration set and red circle represents prediction set.

**Table 1 foods-12-02968-t001:** Detection range, resolution, sampling precision and repeatability of each sensor.

Sensor	Detection Range	Resolution	Precision	Repeatability
C_2_H_4_	0–100 ppm	0.1 ppm	±2% FS	±1% FS
O_2_	0–30% VOL	0.1% VOL	±2% FS	±1% FS
VOC	0–50 ppm	0.001 ppm	±2% FS	±1% FS
CO_2_	0–5000 ppm	1 ppm	±2% FS	±1% FS
Temperature	−20–80 °C	0.1 °C	±0.3 °C	±1% FS
Humidity	0–100% rh	0.1 rh	±0.3% rh	±1% FS

% FS: the percentage of accuracy and full scale.

**Table 2 foods-12-02968-t002:** Prediction model results of days before apple spoilage using C_2_H_4_, CO_2_, VOC and O_2_ sensor data.

Model	Calibration Set		Prediction Set	
	R_c_	RMSEC	R_p_	RMSEP
GA-PLS	0.772	1.481	0.669	1.769
SA-PLS	0.942	0.763	0.936	0.828
ACO-PLS	0.971	0.538	0.926	0.872
CARS-PLS	0.866	1.163	0.859	1.218

**Table 3 foods-12-02968-t003:** Characteristic variables and original variable ranges of each sensor in the apple spoilage early warning model.

Sensor	Characteristic Variables	Original Variable Ranges
Temperature	228, 309	0–500
Humidity	622, 726	500–1000
CO_2_	1064, 1126, 1188	1000–1500
C_2_H_4_	1526, 1538, 1861, 1889, 1894, 1974	1500–2000
O_2_	2001, 2159, 2163, 2274	2000–2500
VOC	2561, 2758, 2965	2500–3000

**Table 4 foods-12-02968-t004:** Independent variables, dependent variables and coefficients of apple spoilage early warning model.

Number	Independent Variables	Dependent Variables	Number	Independent Variables	Dependent Variables
1	0.3264	228	11	−0.0136	1889
2	0.3708	309	12	−0.0118	1894
3	0.0248	622	13	−0.0132	1974
4	0.0363	726	14	0.3407	2001
5	−0.0008	1064	15	−1.9581	2159
6	−0.0005	1126	16	0.3719	2163
7	−0.0014	1188	17	0.5173	2274
8	0.4734	1526	18	−1.9010	2561
9	0.3338	1538	19	0.0013	2758
10	0.0248	1861	20	−0.0009	2965
Coefficient	38.9899				

## Data Availability

The data used to support the findings of this study can be made available by the corresponding author upon request.

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
