# Peer review of "Spoilage Monitoring and Early Warning for Apples in Storage Using Gas Sensors and Chemometrics"

_foods, 2023, doi:10.3390/foods12152968_

Round 1
Reviewer 1 Report
Monitoring the quality of apples is very important. Apples are the most common product bought in the world. The article is correctly written. It contains a thought sequence.
The methodology should include literature references. The discussion should go deeper. More comparisons to other monitoring methods. Apples are extensively covered in the literature, so a more detailed discussion can be added.
Author Response
General comment:
Monitoring the quality of apples is very important. Apples are the most common product bought in the world. The article is correctly written. It contains a thought sequence.
Response: Thank you for the careful reading of the manuscript and the positive comments.
Comment 1:
The methodology should include literature references.
Response: Thank you for the input. The necessary references have been added to the methodology section in the manuscript.
Comment 2:
The discussion should go deeper. More comparisons to other monitoring methods. Apples are extensively covered in the literature, so a more detailed discussion can be added.
Response: We have considered the comment, and the discussion section has been improved to include more work on apple spoilage and monitoring. A comparison was also made to other spoilage prediction models associated with apple quality.
Reviewer 2 Report
The work concerns important problems related to food storage.
My comments:
1. The abstract requires significant proofreading. It should contain: 2-3 introductory sentences, Aim of the work, Clear presentation of the Methodology and Results, and Conclusions.
2. The Methodology and Results chapters require an unambiguous description.
Author Response
General comment:
The work concerns important problems related to food storage.
Response: Thank you very much for your comment. The current work makes a significant advancement in the field of food spoilage monitoring.
Comment 1:
The abstract requires significant proofreading. It should contain: 2-3 introductory sentences, Aim of the work, Clear presentation of the Methodology and Results, and Conclusions.
Response: Your comment is much appreciated. The abstract has been changed to include introductory sentences, the aim of the work, methodology and results.
Comment 2:
The Methodology and Results chapters require an unambiguous description.
Response: Thank you very much for the input. An additional section has been provided at the beginning of the methodology section to improve the understanding of the experimental methods carried out. An additional explanation has been given in the result section to clarify the obtained results.

Reviewer 3 Report
1) Good article
2) I've no issue with giving this paper to be published
3) The dataset is a good representative of the IoT 4.0IR. Could this be synced as part of the cloud data so that it can be accessed remotely?
4) Please include the possibility of having 4.0IR element in the studies. At what stage should it be included?
5) Could this be part of the SDG in food security elements? How? Discuss it.
Author Response
General comment:
Good article. I've no issue with giving this paper to be published
Response: Thank you very much for your positive comment.
Comment 1:
The dataset is a good representative of the IoT 4.0IR. Could this be synced as part of the cloud data so that it can be accessed remotely?
Response: We appreciate your feedback. The dataset is a good representative of the IoT 4.0IR because it uses a variety of sensors to collect data about the environment, and it uses a machine learning model to predict spoilage. This type of data is becoming increasingly important as the IoT continues to grow. Syncing the dataset to the cloud would make it more accessible to researchers and other interested parties. It would also make it easier to share the data with others, which could help to advance the field of IoT research.
Comment 2:
Please include the possibility of having 4.0IR element in the studies. At what stage should it be included?
Response: Thank you very much for an insightful comment. The research employs several elements of the 4.0IR, including cyber-physical systems, the IoT, and big data. The gas monitoring array employed in the research is associated with a cyber-physical system, that combines physical sensors with digital components to create a smart monitoring prototype that is capable of sensing and analyzing the environment. The monitoring prototype is connected to the IoT, which allows the collection and exchange of data with other devices. Additionally, big data comes into play in analyzing the data collected by the gas monitoring array. This data can then be used to identify patterns and trends that can be used to predict spoilage.
Comment 3:
Could this be part of the SDG in food security elements? How? Discuss it.
Response: Thank you very much for your input. The study could be a part of the SDG in food security. Goal 2 of the SDGs is to "end hunger, achieve food security and improved nutrition and promote sustainable agriculture". The current work involves the development of a system for monitoring and early warning of apple spoilage. This system could be used to prevent food waste, which is a major problem and contributes to hunger and malnutrition. By monitoring the environment in which apples are stored and providing early warning of spoilage, necessary action could be taken to prevent further spoilage and post-harvest loss during storage. The system could also be used to improve food safety, which is another important goal of the SDGs. The system could help to improve food safety by monitoring the environment in which apples are stored and identifying potential risks of foodborne illness. By monitoring the levels of gas components in the storage environment, the system could identify potential risks of foodborne illness even before the bacteria or other microorganisms have grown to harmful levels. Thus, early detection could then be used to take action to mitigate the risks of foodborne illness.

Reviewer 4 Report
The authors have worked on the hardware prototype development for the early spoilage detection of apples. The work methodology and results are are not definitive with current scientific standards, and the novelty of the work is speculative. Especially when the authors have incorporated different senors, which are already being used in storage facilities for monitoring of fruit spoilage. The main theme of the manuscript seems to be data collection and processing with different mathematical and statistical models, which has already been studied by many other authors.
1) Authors needs to point out what is the novelty of their study? Whether this study adds significant insight in to the state of art development in food spoilage monitoring.
2. Images of apples on different storage days has to be given, especially for apples inoculated with spoilage microbe
3. The microbial growth data is missing in the work, Authors have not mentioned about any enumeration of spoilage microbes in apples at different storage days.
4. What criteria was followed to detect the acceptable limit of apple spoilage? The acceptable limit for each sensor readings were not mentioned anywhere in the study. Simple correlation of data with different mathematical equation cannot give the acceptable limit of apple spoilage. With this drawback, the work seems to be partial rather than complete.
5. If authors are to inoculate the apples with spoilage microbes, the microbe has to be enumerated periodically and this data has to be correlated with the other sensor data to provide some new insights in prediction of apple spoilage. The reviewer would like to urge the authors to correlate the analysis values with acceptable limits to give exact shelf life/spoilage predictions.
Author Response
General comment:
The authors have worked on the hardware prototype development for the early spoilage detection of apples. The work methodology and results are not definitive with current scientific standards, and the novelty of the work is speculative. Especially when the authors have incorporated different sensors, which are already being used in storage facilities for monitoring of fruit spoilage. The main theme of the manuscript seems to be data collection and processing with different mathematical and statistical models, which has already been studied by many other authors.
Response: We appreciate your feedback. The main objective of the work is to develop a monitoring prototype that can be employed to predict and visualize apple spoilage from gas composition analysis. The prototype developed in this study is a valuable starting point for future research for developing an early warning system for food spoilage.
Comment 1:
Authors needs to point out what is the novelty of their study? Whether this study adds significant insight in to the state of art development in food spoilage monitoring.
Response: Thank you for the feedback. The study developed a remote monitoring and early warning platform that can be used to visualize the apple warehouse's sensors data and spoilage level. This platform is a valuable tool for farmers and distributors, as it allows them to monitor the condition of their apples in real-time and to take action to prevent spoilage. The study also developed a multi-factor fusion early warning model based on various modelling methods. This model was able to predict apple spoilage with a high degree of accuracy. This suggests that it is possible to develop early warning systems that are based on a gas component measured during storage. The model was also able to identify the relevant variables in the vast data set collected from the sensors removing the redundant information which made the model more accurate for the prediction of apple spoilage. In summary, the study makes a significant contribution to the state of the art in food spoilage monitoring. It proposed a novel approach to detect spoilage using sensor data, and it developed a multi-factor fusion early warning model to predict spoilage with a high degree of accuracy. The study also developed a remote monitoring and early warning platform that is a valuable tool for farmers and distributors.
Comment 2:
Images of apples on different storage days has to be given, especially for apples inoculated with spoilage microbe
Response: Thank you for the valuable input. Images of apples with different degrees of spoilage have been added to Figure 3 in the manuscript.
Comment 3:
The microbial growth data is missing in the work, Authors have not mentioned about any enumeration of spoilage microbes in apples at different storage days.
Response: Thank you for the input. The developed method is not an alternative technique to measure the microbial growth in apples during storage. Instead, the system is designed to predict the spoilage levels of apples by monitoring the sensor readings during storage. Thus, microbial enumeration was not considered to be a critical component of this system.
Comment 4:
What criteria was followed to detect the acceptable limit of apple spoilage? The acceptable limit for each sensor reading was not mentioned anywhere in the study. Simple correlation of data with different mathematical equations cannot give the acceptable limit of apple spoilage. With this drawback, the work seems to be partial rather than complete.
Response: Thank you very much for your comment. The multi-factor fusion early warning model was created by considering data from all the sensors. This means that the model is not dependent on any one particular sensor data. If there was an acceptable limit for each sensor, the model would be more likely to give misleading results. Thus, it is important to consider data from all the sensors when determining the acceptable limit for apple spoilage. By considering data from all the sensors, the model can take into account the relationships between the different sensor readings and the model is less likely to be misled by changes in a single sensor reading.
Comment 5:
If authors are to inoculate the apples with spoilage microbes, the microbe has to be enumerated periodically and this data has to be correlated with the other sensor data to provide some new insights in prediction of apple spoilage. The reviewer would like to urge the authors to correlate the analysis values with acceptable limits to give exact shelf life/spoilage predictions.
Response: We appreciate your feedback and will take it into consideration. However, this was not the aim of our study. The aim of our study was to develop a sensor prototype that is capable of monitoring the volatile gas production to predict spoilage of apples. We believe that this sensor prototype has the potential to be used to develop an early warning system for apple spoilage. We agree that the correlation analysis with colony counting data will provide more accurate shelf life/spoilage predictions. However, this is beyond the scope of our study and the authors hope that future studies will explore this further.

Round 2
Reviewer 2 Report
I have no comments
Reviewer 4 Report
The authors have addressed all the comments given by the reviewer, thus the manuscript may be considered for publication. But the reviewer is still not sure about the scope of this manuscript for publication in journal of foods. Thus it is left to the discretion of the editor.